# Patient preferences in the treatment of hemophilia A: A latent class analysis

**Axel C. Mühlbacher**[1,2,3]*, **Andrew Sadler**[1], **Björn Lamprecht**[4], **Christin Juhnke**[1]

**1** Health Economics and Health Care Management, Hochschule Neubrandenburg, Neubrandenburg, Germany, **2** Gesellschaft für empirische Beratung GmbH, Freiburg, Germany, **3** Duke Department of Population Health Sciences and Duke Global Health Institute, Duke University, Durham, North Carolina, United States of America, **4** Roche Pharma Aktiengesellschaft, Grenzach-Wyhlen, Germany

\* muehlbacher@hs-nb.de

## Abstract

### Objective

To examine subgroup-specific treatment preferences and characteristics of patients with hemophilia A.

### Methods

Best–Worst Scaling (BWS) Case 3 (four attributes: application type; bleeding frequencies/ year; inhibitor development risk; thromboembolic events of hemophilia A treatment risk) conducted via online survey. Respondents chose the best and the worst option of three treatment alternatives. Data were analyzed via latent class model (LCM), allowing capture of heterogeneity in the sample. Respondents were grouped into a predefined number of classes with distinct preferences.

### Results

The final dataset contained 57 respondents. LCM analysis segmented the sample into two classes with heterogeneous preferences. Preferences within each were homogeneous. For class 1, the most decisive factor was bleeding frequency/year. Respondents seemed to focus mainly on this in their choice decisions. With some distance, inhibitor development was the second most important. The remaining attributes were of far less importance for respondents in this class. Respondents in class 2 based their choice decisions primarily on inhibitor development, also followed, by some distance, the second most important attribute bleeding frequency/year. There was statistical significance ($P < 0.05$) between the number of annual bleedings and the probability of class membership.

### Conclusions

The LCM analysis addresses heterogeneity in respondents' choice decisions, which helps to tailor treatment alternatives to individual needs. Study results support clinical and allocative decision-making and improve the quality of interpretation of clinical data.

**Data Availability Statement:** The datasets generated and/or analyzed during the current study are not publicly available due to EU General Data Protection Regulation. Data are available for sharing after reasonable request and conforming to

the use in health preference research (HPR) for noncommercial aims. Roche data protection policies apply. Although the authors cannot make their study's data publicly available at the time of publication, Roche commits to make the data underlying the findings described in this study fully available without restriction to those who request the data, in compliance with the Roche Data policy. For data sets involving personally identifiable information or other sensitive data, data sharing is contingent on the data being handled appropriately by the data requester and in accordance with all applicable local requirements. Request to access study data should be sent to: grenzach.biometrics-hub-europe-africa@roche.com.

**Funding:** This study was financed by Roche Pharma AG (https://www.roche.de). Employees of the sponsor are listed as authors and were involved in the study design, data collection and analysis, decision to publish, or preparation of the manuscript. All authors received support for third-party editing assistance, provided by Roche Pharma AG, Grenzach-Wyhlen, Germany or GEB mbH, Germany. Prof. Dr. Mühlbacher is employed at Hochschule Neubrandenburg and GEB mbH and reports grants for studies, presentations and advisory boards (national and international) from Abbott, AbbVie, Actelion, Amgen, AstraZeneca, Baxter, Bayer, Boehringer Ingelheim, BPI, Bristol Myers Squibb, Credopard GmbH, Daiichi Sankyo, Eli Lilly, Genentech, Gilead Sciences, GlaxoSmithKline, Grünenthal, Insight Health, IQWiG, Ipsen Pharma, Janssen-Cilag, Johnson & Johnson, Merck, MSD, Merck KGaA, NICE, Novartis, Novo Nordisk, Pfizer, Roche, Sanofi-Aventis, Stallergenes, Shire, Steigerwald Arzneimittelwerk, VFA, ViiV Healthcare GmbH, Wyeth. Drs. Juhnke and Sadler are employed at Hochschule Neubrandenburg and report grants from GEB mbH during the conduct of the study. Dr. Lamprecht is an employee of Roche Pharma AG, Grenzach-Wyhlen, Germany. This does not alter our adherence to PLOS ONE policies on sharing data and materials. The funder provided support in the form of a salary for BL, but did not have any additional role in the study design, data collection and analysis, decision to publish, or preparation of the manuscript. The specific role of BL is articulated in the 'Author contributions' section.

**Competing interests:** All authors received support for third-party editing assistance, provided by Roche Pharma AG, Grenzach-Wyhlen, Germany or GEB mbH, Germany. Prof. Dr. Mühlbacher is employed at Hochschule Neubrandenburg and GEB mbH and reports grants for studies, presentations and advisory boards (national and

## Introduction

Hemophilia, an inherited, rare bleeding disorder, is complex to diagnose and manage [1]. Hemophilia A is more common than B; ~8/10 people with hemophilia have type A [2]. Patients suffer from repeated hemorrhagic episodes in joints and soft tissues [3]. Patients bleed longer than other people, and bleedings can occur internally in joints and muscles, as well as externally via minor cuts, dental procedures, or trauma [4].

An increasing number of people with bleeding disorders have been identified since 1999. The World Federation of Hemophilia identified 111,203 people in 1999 with inherited bleeding disorders such as hemophilia and 337,641 in 2018 [5]. The number of hemophilia cases was 78,629 in 1999 and 210,454 in 2018. Here, the number of 173,711 hemophilia A cases accounted for about 83% of the total hemophilia cases in 2018. The proportion of hemophilia A in the total number of hemophilia cases was 53,864 in 1999 and 173,711 in 2018, which corresponds to an increase of over 200%. In Germany, medical data of patients with bleeding disorders are consolidated in the German Hemophilia Registry (Deutsches Hämophilieregister, DHR). In 2018, 4,240 people with hemophilia A and 785 people with hemophilia B were included in the registry and 2,583 (~61%) of hemophilia A and 403 (~51%) of hemophilia B cases had severe hemophilia [6].

In a first paper we analyzed patients' preferences of a whole sample set regarding general relative importance of all attributes with a mixed logit model [7]. Here, we aimed to assess heterogeneity of patients' preferences for alternative hemophilia A treatments in Germany; the main focus being to analyze possible differences in preference patterns in the sample regarding treatment characteristics. A Best–Worst Scaling (BWS) Case 3 was conducted in an interviewer-administered survey. Two classes of respondents with heterogeneous preferences were identified in the current latent class model analysis.

## Methods

### Ethics statement

Participants were recruited by an external market research company. Prior to participation, patients gave informed consent. Participation was voluntary and anonymous [7]. All documents used in the study were primarily reviewed by the ethics committee at the State Medical Association of Baden-Württemberg. The study was declared harmless and approved (F-2017-048). In addition, the study was notified to the State Medical Associations of Hessen, Lower Saxony and Saarland, as these are the chambers responsible for the study centers where patient recruitment took place. This study used an anonymous online data collection. All patients gave a virtual declaration of consent before the actual survey by clicking a corresponding box and thus agreed to participate in the research project. All participants were comprehensively informed and enlightened about the research project. The patients were free to answer the questions addressed to them or to terminate the questionnaire at any time during the survey. No IP addresses of the computers, tablets etc. used or other security relevant information was collected.

### Best–worst scaling case 3

Discrete choice experiments (DCE) are quantitative methods for measuring stated preferences. They are widely accepted in healthcare, and increasingly discussed by regulatory bodies [8–11]. DCEs assume a health product/service can be described in terms of its attributes and levels of its attributes. The attributes' relative importances are determined by analyzing tradeoffs between decision-relevant attributes and their levels. DCE participants choose between ≥2

international) from Abbott, AbbVie, Actelion, Amgen, AstraZeneca, Baxter, Bayer, Boehringer Ingelheim, BPI, Bristol Myers Squibb, Credopard GmbH, Daiichi Sankyo, Eli Lilly, Genentech, Gilead Sciences, GlaxoSmithKline, Grünenthal, Insight Health, IQWiG, Ipsen Pharma, Janssen-Cilag, Johnson & Johnson, Merck, MSD, Merck KGaA, NICE, Novartis, Novo Nordisk, Pfizer, Roche, Sanofi-Aventis, Stallergenes, Shire, Steigerwald Arzneimittelwerk, VFA, ViiV Healthcare GmbH, Wyeth. Drs. Juhnke and Sadler are employed at Hochschule Neubrandenburg and report grants from GEB mbH during the conduct of the study. Dr. Lamprecht is an employee of Roche Pharma AG, Grenzach-Wyhlen, Germany. This does not alter our adherence to PLOS ONE policies on sharing data and materials. The funder provided support in the form of a salary for BL, but did not have any additional role in the study design, data collection and analysis, decision to publish, or preparation of the manuscript. The specific role of BL is articulated in the 'Author contributions' section

hypothetical alternatives characterized by different attributes in repeated choice scenarios. Assuming rational choice decisions, the treatment alternative benefit and its characteristics can be analytically determined [12].

A BWS is a special form of classical DCE. In a survey using a BWS Case 3 (multi-profile case) approach there are ≥3 alternatives in a choice scenario, and respondents are required to identify the most and least preferred alternative in each. Assuming there are only three alternatives in a choice task, a full ranking of preferred alternatives can be obtained. In case of >3 alternatives, a follow-up question regarding the remaining alternatives between the previously chosen best and worst alternative is answered. The BWS is approved as a valuable way to analyze patients' preferences [13–16].

## Attributes and levels

To identify treatment attributes for the BWS survey, a literature search and qualitative pre-test interviews with 12 patients were conducted [7]. Based on these, four attributes with various levels were chosen: type of application (intravenous/subcutaneous application: 1/2/3x per week); development of inhibitors (no [0%]/ low [2%]/ medium [4%]); bleeding frequency/year (0/5/15/25 per year); risk of thromboembolic events (no [0%]/ low [1%]/ medium [2%] risk [7]).

## Experimental design

The combination of attribute levels resulted in an experimental design ($3^2$ x $4^1$ x $6^1$; two attributes with three levels, one attribute with four, and one attribute with six) with 216 possible choice alternatives [7]. A d-efficient fractional factorial design with 240 choice scenarios was created with Ngene version 1.2.1 (ChoiceMetrics, Sydney, NSW) [17]. A dominance test was used to assess validity (rationality) of respondents' choice decisions. Within a choice scenario, containing three treatment alternatives, respondents were asked to choose the most and least preferable alternative. Attributes were randomized to control for order effects across respondents. S1 Fig shows a screenshot of the example choice set.

## Recruitment

The study has been conducted in two phases. Prior to the main survey, a qualitative preliminary study was conducted using qualitative interviews with affected patients (N = 12). After evaluation of these interviews, the main survey was conducted from October 2018 to May 2019. The participants were recruited in cooperation with an independent market research company, specialized in the recruitment of target groups in the healthcare sector. Patients with hemophilia A patients who were at least 18 years old, had adequate German language skills, and agreed to participate in the 30- to 60-minute study interview were eligible for the survey [7]. The patient survey was conducted by interview and with the aid of a computer-assisted questionnaire. The survey was accessed via a provided link. 94.7% of respondents were male, average age was 34.68 years (standard deviation: 13.5). For more information on the sample characteristics please see S1 Table.

## Statistical analysis

A latent class model (LCM) was used to analyze the data [18, 19]. LCMs assume that respondents differ in terms of their preferences and that individual choice behavior depends on observable attributes and on latent heterogeneity that varies with unobservable factors. Individuals are implicitly sorted into a set of different latent classes with homogeneous preferences.

The classes are latent, since it is not determined a priori which respondent belongs to which class. LCMs are semi-parametric models, which free the analyst from unwarranted distributional assumptions. The models have a discrete distribution of coefficients and are used to uncover possible different preference pattern among classes. Class membership is represented in terms of class probability, which may depend on socio-economic characteristics of the respondent, general attitudes, previous treatment experience or other factors. LCM approaches offer insights into the heterogeneity of patient preferences that are not readily identifiable through other discrete choice models, especially when there are reasons to believe that these preferences are clustered around certain values [20].

The aim of the LCM was to quantify the average impact of attribute levels on the therapy preference. For each attribute level, both the mean coefficient, standard error and the 95% confidence interval were estimated, and for each attribute the relative importance. The attribute level with the highest coefficient in each class is most decisive in the choice of a therapy. The relative attribute importance for each class was calculated using the difference between the largest and the smallest level coefficients for each attribute. A score of 10 was given to the attribute with the largest difference. All other importance scores were calculated in relation to the most important attribute. Effects coding was used for the attributes in the analysis, where the reference level is generated by the inverted sum of the remaining levels.

Analysis of the LCM was conducted with Stata 15.1 (StataCorp, College Station, TX).

Three respondents failed the rationality test using dominant choice alternatives. The analysis with and without these respondents yielded better results for the model without the failed respondents. Eventually, the data of 57 respondents were used for the final analysis.

## Results

### Sample population

The overall sample included 57 respondents [7]. Given the small size, two classes were used for the LCM analysis. The probability that a respondent would be assigned to a particular class depended on the sequence of respondents' choice decisions. Average probability of class membership was 0.9884 for class 1, and 0.9698 for class 2. There was no uncertainty regarding assignment of respondents to one of the classes. Respondents were more likely to be assigned to class 1. Class 1 included 65% of the sample and class 2 included 35%. There were no statistically significant differences between classes' sociodemographic characteristics (S2 Table for Class 1/2 and [7] for total).

### LCM: Capturing heterogeneity

The two classes differed in their preferences regarding treatment of hemophilia A. Table 1 shows the latent class analysis results. The table includes the mean coefficients (utilities) along with corresponding standard errors, confidence intervals, and significance levels. Coefficients are scaled to sum to zero within each attribute, and are interpreted as relative utility of each attribute level. Higher values are associated with higher preferences. Attribute levels with positive signs were preferred over levels with negative signs by the respondents in the choice decisions. The larger the absolute value of a coefficient, the greater the impact on respondents' choice decisions. Large negative coefficients indicate a large negative impact on choice decisions. Across classes, the signs of all coefficients were as expected. A less frequent application was preferred to a more frequent one, a smaller number of bleedings was preferred to a higher number, and a lower risk of inhibitor development and thromboembolic events was preferred to a higher risk. However, there were differences between the classes in the magnitude of the impact of the attribute and levels.

**Table 1. Results of the latent class analysis.**

| | Class 1 (65%) | | | | | Class 2 (35%) | | | | |
|---|---|---|---|---|---|---|---|---|---|---|
| Level | Coef. | SE | | 95% CI | Sig | Coef. | SE | | 95% CI | Sig |
| *Type of application* | | | | | | | | | | |
| Intravenous 1x/week | 0.44 | 0.14 | 0.17 | 0.72 | *** | 0.28 | 0.19 | -0.10 | 0.65 | |
| Intravenous 2x/week | 0.03 | 0.13 | -0.21 | 0.28 | | -0.02 | 0.20 | -0.41 | 0.36 | |
| Intravenous 3x/week | -0.44 | 0.14 | -0.72 | -0.16 | *** | -0.84 | 0.21 | -1.24 | -0.44 | *** |
| Subcutaneous 1x/week | 0.17 | 0.15 | -0.12 | 0.46 | | 1.02 | 0.21 | 0.61 | 1.42 | *** |
| Subcutaneous 2x/week | 0.26 | 0.12 | 0.02 | 0.51 | ** | -0.17 | 0.20 | -0.56 | 0.23 | |
| Subcutaneous 3x/week | -0.46 | 0.14 | -0.74 | -0.19 | *** | -0.26 | 0.19 | -0.63 | 0.11 | |
| *Development of inhibitors* | | | | | | | | | | |
| No (0%) | 1.13 | 0.11 | 0.92 | 1.35 | *** | 2.92 | 0.22 | 2.48 | 3.35 | *** |
| Low (2%) | 0.13 | 0.08 | -0.03 | 0.30 | | 0.46 | 0.14 | 0.18 | 0.73 | *** |
| Medium (4%) | -1.27 | 0.11 | -1.48 | -1.05 | *** | -3.37 | 0.27 | -3.90 | -2.84 | *** |
| *Bleeding frequency per year* | | | | | | | | | | |
| 0 bleedings | 2.91 | 0.17 | 2.59 | 3.24 | *** | 0.89 | 0.19 | 0.52 | 1.26 | *** |
| 5 bleedings | 1.55 | 0.12 | 1.31 | 1.79 | *** | 0.83 | 0.15 | 0.53 | 1.13 | *** |
| 15 bleedings | -0.95 | 0.12 | -1.18 | -0.73 | *** | -0.25 | 0.15 | -0.55 | 0.04 | * |
| 25 bleedings | -3.51 | 0.20 | -3.91 | -3.11 | *** | -1.47 | 0.22 | -1.90 | -1.03 | *** |
| *Risk of thromboembolic events* | | | | | | | | | | |
| No risk (0%) | 0.55 | 0.09 | 0.36 | 0.73 | *** | 0.81 | 0.14 | 0.55 | 1.08 | *** |
| Low risk (1%) | 0.13 | 0.08 | -0.03 | 0.30 | | 0.15 | 0.11 | -0.08 | 0.37 | |
| Medium risk (2%) | -0.68 | 0.09 | -0.86 | -0.50 | *** | -0.96 | 0.14 | -1.24 | -0.69 | *** |
| _cons | 0.63 | 0.29 | 0.05 | 1.20 | ** | | | | | |

AIC = Akaike Information Criteria, BIC = Bayesian Information Criterion, CI = confidence interval, Coef. = coefficient, SE = standard error, Sig = significance.

*** $P < 0.01$

** $P < 0.05$

* $P < 0.1$.

ll(model): -720.766; AIC: 1491.532; BIC: 1649.525

It is noticeable that each class has one attribute with very high coefficients: "Bleeding frequency per year" for class 1 and development of inhibitors for class 2. Respondents in class 1 seemed to pay more attention to bleeding frequencies in their choice decisions. In contrast, respondents in class 2 seemed to be more focused on development of inhibitors. Compared with the other attributes within the class, these attributes have very high values for level coefficients for the first and last level. Class 1 respondents showed a strong preference for "0 bleedings" (coef. = 2.91; $P < 0.01$) and strongly disfavored "25" (coef. = –3.51; $P < 0.01$). Class 2 respondents clearly preferred "no (0%)" development of inhibitors (coef. = 2.92; $P < 0.01$), and disfavored "medium" (4%) development (coef. = –3.37; $P < 0.01$). All levels of the respective attributes were statistically highly significant. The large coefficients indicate that respondents of each class seemed to be very confident in their choice decisions regarding these attribute levels.

Another noticeable difference concerns preferences for type of application. While class 1 respondents seemed to prefer weekly intravenous applications (coef: 0.44; $P < 0.01$), class 2 respondents seemed to prefer weekly subcutaneous application (coef: 1.02; $P < 0.01$). Both classes agree that they would reject a three-times-a-week intravenous application. The objection to this application seemed to be even greater for class 2 respondents (coef: –0.84; $P < 0.01$).

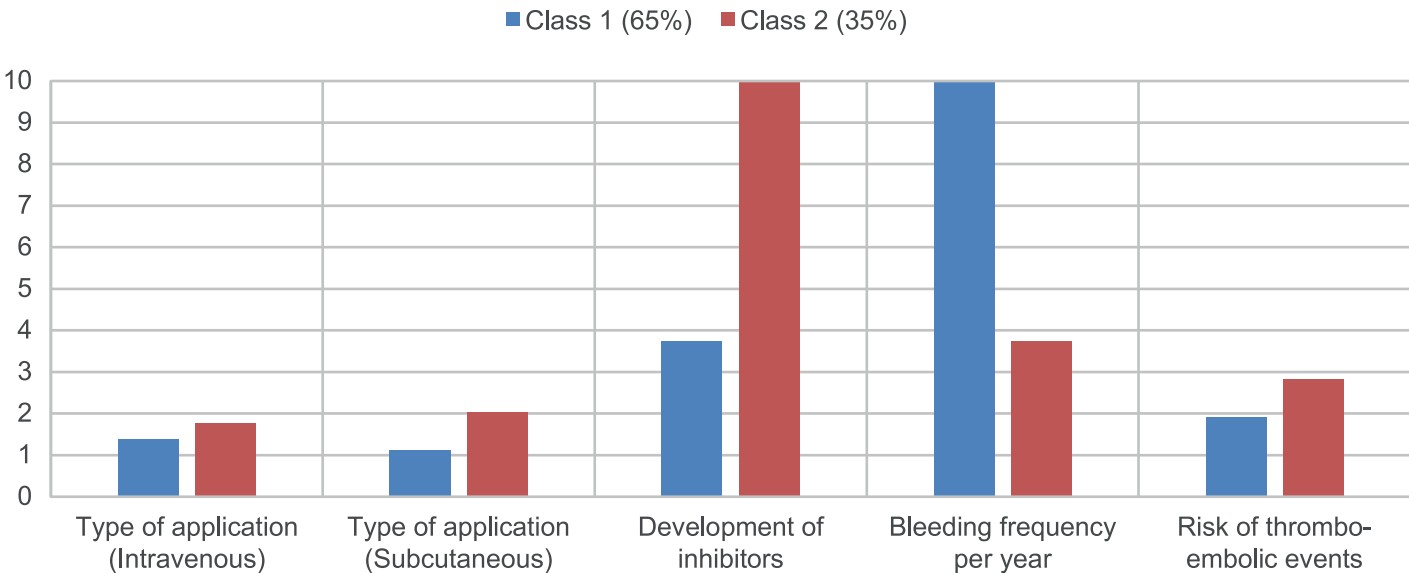

**Fig 1. Relative attribute importance.** The importance of each attribute is derived by the range between the lowest level coefficient and the highest level coefficient. This range is then normalized on a scale of 1 to 10, where 10 is the highest value and thus the most important or preferred attribute. Class 1 respondents mainly focused on bleeding frequency, while class 2 respondents paid more attention to "Development of inhibitors" in choice decisions.

S2 Fig shows a graphical representation of the coefficients of the LCM.

The differences in preferences become clear in Fig 1, which shows the relative importance and illustrates the focus of respondents in choice decisions (type of application was divided into "subcutaneous" and "intravenous"). Class 1 respondents have a greater preference for weekly intravenous application, whereas class 2 respondents prefer a weekly subcutaneous over intravenous application. However, the distance between weekly intravenous and subcutaneous application is smaller for respondents in class 1. These respondents are indifferent between a twice-a-week intravenous and subcutaneous application, and between a three-times-a-week intravenous and subcutaneous application. The respective coefficients are close to each other. "Development of inhibitors" had a larger impact on choice decisions of class 2 respondents. With class 2 respondents the distance from the worst level (medium [4%]) to the best level (no [0%]) is more than twice as large as class 1 respondents. It is the other way around with "Bleeding frequency per year". Table 1 shows, that the change from the most preferred level ("0 bleedings") to the second most ("5 bleedings") is not decisive for the respondents in class 2. The confidence intervals for the two-level coefficients overlap. For class 1, however, the level coefficients are statistically different from each other. The gradient of the line is steeper for class 1 and flatter for class 2. This indicates that a change from one level to the other is more important for class 1 respondents. With regard to the last attribute, the two classes have similar preferences.

## Experiences and probability of class membership

Patients were asked about the characteristics of their therapies and experiences with the disease (S1 Table). Compared with class 2, there was a larger proportion of respondents in class 1 with 0 bleedings in the last year; 27.0% of the respondents in class 1 had 0 bleedings versus only 5.0% in class 2. 43.2% of class 1 and 65.0% of class 2 indicated that maximum number of bleedings was >10. Statistical significance was identified between the severity of patients' hemophilia and class membership, and between current state of health and class membership

(P < 0.05). 78.4% of class 1 and 60.0% of class 2 suffered from severe hemophilia. Only 8.1% of class 1 suffered from moderate disease, while 40.0% of class 2 had moderate hemophilia. Mild disease only applied to class 1 (13.5%). The question about current health status (P < 0.05) was answered by 18.9% respondents of class 1 and 50.0% of class 2 with "very good", and by 62.2% of class 1 and 20.0% of class 2 with "good". Most respondents in class 1 (81.1%) regularly administered their current therapy. Conversely, in class 2 60.0% followed a regular drug administration schedule while 40.0% took the drugs only on demand. The majority of class 1 respondents (91.9%) had a treatment plan that provided intravenous administration. In class 2, 80.0% of the respondents used intravenous administration. Detailed cross tables are shown in S3 Table, along with cross tables to compare the latent classes and another LCM with independent variables to identify characteristics of class membership. In the LCM analysis, additional covariates were included as class-membership effects for the two classes in the model. Variables that showed a significant or an almost-significant level in the cross tables were tested as covariates in the LCM. These variables were binary and assumed to be constant across alternatives for the same respondent. Only three covariates which showed significance of at least P < 0.05, were included in the final model (S3 Table). The first covariate (cov1) was derived from the question "How many bleedings have you had in the last year?" and coded 1 if respondents had 0–2 bleedings and 0 otherwise. cov2 was derived from "What was the maximum number of bleedings?" and coded 1 if respondents stated that they suffered from >20, and 0 otherwise. cov3 was based on "How would you describe your current state of health in general?" and coded 1 if respondents answered with "very good" and 0 otherwise. Reference class is class 2. Respondents in class 1 differed significantly in terms of bleeding frequencies, maximum number of bleedings, and current state of health. Class 1 respondents had a lower number of bleedings in the last year (coef. = 2.04; P < 0.05). Respondents of class 2 were more likely to have a maximum number of bleedings of >20 (coef. = –2.56; P < 0.01). Regarding current state of health, class 1 had a significantly lower proportion of respondents with a very good health state (self-report by respondents) (coef. = –2.91; P < 0.01) than class 2.

## Discussion

People often form groups or segments, also latent classes, with similar interests and needs and seek similar benefits from health providers. Health organizations need to understand whether the same health treatments, prevention programs, services, and products should be applied to everyone in the relevant population or whether different treatments need to be provided to each of several groups that are relatively homogeneous internally but heterogeneous among groups. Using panel data from discrete choice experiments, latent class analysis is commonly performed to identify subsets of participants with homogeneous preferences within groups and heterogeneous preferences between groups [21]. Classification of respondents into groups or clusters is usually based on the patterns of outcome variables such as individual choice decisions in stated-preference surveys using discrete choice experiments [22].

Hemophilia A is a severe, chronic disease that makes high demands of patients' treatment compliance. Providing therapy is very complex and requires a balance between possible benefits and risks. It is important to know the preferences and wishes of patients with regard to the treatment alternatives and application schemes of the drugs. The practical approach provided by the BWS can help improve communication between patients and their care providers, support clinical and allocative decision-making, and improve the quality of interpretation of clinical data [7]. Therapies can also be made to be more patient-oriented based on the knowledge gained from this method. As such, care can be made more effective and efficient; increasing benefits for patients [7]. Two classes of respondents with different preferences were identified

in the current LCM analysis. Overall, results showed that patients attach the highest importance to "Bleeding frequency per year" and risk of "Development of inhibitors". A higher frequency of bleeding and a higher risk of inhibitor development would significantly impact patients' choice decisions. Class 1 respondents made their choice decisions mainly with focus on "Bleeding frequency per year" while class 2 respondents paid more attention to "Development of inhibitors". The "Type of application" did not seem to influence the choice decision much compared with other attributes. In this respect, however, there was a difference between the two classes. Regarding class characteristics, the two seemed to be segmented in terms of experiences with the disease. The classes could be described in terms of patients' experience with the disease and current state of health.

Heterogeneous preference structures may influence the acceptance, and thus the adherence, of patients to alternative therapies. Socio-demographic variables, experiences and attitudes can cause heterogeneous preference structures. With latent class analysis, these preference structures can be revealed and divided into segments. The distribution of patients in both latent classes is tabulated (see S2 Table). Comparison of the sociodemographic characteristics of the patients in both latent classes by chi-square tests revealed no significant differences between class members. However, there were significant differences between class members regarding treatment experience, number of bleeds, maximum number of bleeds, type of current treatment, severity of hemophilia A, and self-reported health status. In our analysis, we found that, compared with class 1, class 2 contained a significantly larger proportion of patients with more than 2 bleeding episodes per year. In addition, there were more patients in this class who had experienced a maximum number of bleeding episodes greater than 20. For these patients in class 2, avoiding the development of inhibitors was the most decisive criterion when choosing a therapy alternative. Patients who had more bleeds per year placed greater weight on the attribute of development of inhibitors when choosing a therapy. Avoidance of inhibitors was more important for patients in class than for patients in class 1. Based on this information, physicians can communicate and target therapy.

## Limitations

A limitation is the small sample size. Due to the rare nature of the disease and the low prevalence in Germany, it was difficult to recruit patients to achieve a larger size. The sample of N = 57 and therefore the sizes of the latent classes tend to be towards the lower limit. This is also reflected in the high standard errors. A larger sample size is recommended for LCM; most studies in other indications include >300 respondents [21].

There are two types of heterogeneity in discrete choice data. One type is called taste (or preference) heterogeneity, and the other is called scale heterogeneity. Taste heterogeneity identifies groups of respondents who like or dislike different alternatives in a systematic and quantifiable way. Scale heterogeneity refers to the error variance which is thought to vary systematically in response to task complexity, e.g., the number of choice alternatives. That is, heterogeneity cannot be linked to measurable aspects of the decision maker. Scale heterogeneity can be an issue, particularly when comparing coefficients from datasets from different populations or data generated from different sources, e.g., sources using different sampling strategies [23–25]. As we did not intend to analyze how patients made a decision, but rather what kind of decision they made, in our study we did not separate preference heterogeneity from scale heterogeneity. The main interest of the study was to investigate systematic preference heterogeneities, e.g., whether there are differences in therapy uptake in the group of hemophilia patients and which therapy characteristics are accounting for the difference. Since we examine a very small and homogeneous patient population in the study, the analysis of scale heterogeneity was omitted.

## Conclusions

This study analyzed patients' preferences for hemophilia A treatments using a BWS. Data were analyzed using the LCM approach, which addresses heterogeneity in respondents' choice decisions. Study results identified two classes of respondents with different preferences: respondents of class 1 made their choice decisions mainly with a focus on the attribute "Bleeding frequency per year," while respondents of class 2 paid more attention to "Development of inhibitors." This helps in decision-making to tailor treatment alternatives for hemophilia A patients to individual needs.

## Supporting information

**S1 Fig. Screenshot of the example choice set.** The question is: "You are diagnosed with hemophilia A. The doctor asks you to choose between therapy A, therapy B and therapy C. In your opinion, which therapy is the best and which is the worst?".
(DOCX)

**S2 Fig. Coefficients of the latent class model.** Coefficients are displayed separately for each of the two classes. Vertical bars around the coefficients represent the 95% confidence interval. When the confidence intervals overlap for adjacent levels within an attribute, the coefficients of these levels are statistically not different from each other. The upper graphic shows coefficients of class 1 respondents, the lower graphic coefficients of class 2 respondents. The first attribute shows the two types of application, intravenous and subcutaneous, on top of each other. Also, the greater distance between the values "5 bleedings" and "15 bleedings" as well as between "15 bleedings" and "25 bleedings" compared with the distance between 0 and 5 bleedings of the attribute "Bleeding frequency per year" is taken into account in the graph.
(DOCX)

**S1 Table. Characteristics of the therapy and experiences.** * $P < 0.1$, ** $P < 0.05$, *** $P < 0.01$.
(DOCX)

**S2 Table. Sociodemographic characteristics of the overall sample and the two latent classes.**
(DOCX)

**S3 Table. Latent class model with membership variables.** CI = confidence interval, Coef. = coefficient, LCM = latent class model, SE = standard error, Sig = significance. *** $P < 0.01$, ** $P < 0.05$, * $P < 0.1$. ll(model): -709.411; AIC: 1474.821; BIC: 1651.773; degrees of freedom: 28. In the LCM analysis additional covariates were included as class-membership effects for the two classes in the model. Variables that showed a significant or an almost-significant level in the cross tables (see S1 Table) were tested as covariates in the latent class model. These variables were binary and assumed to be constant across alternatives for the same respondent. Only three covariates which showed a significance of at least $P < 0.05$ were included in the final model. The first covariate (cov1) was derived from the question "How many bleedings have you had in the last year?" and coded 1 if respondents had 0 to 2 bleedings last year and 0 otherwise. Second covariate (cov2) was derived from the question "What was the maximum number of bleedings?" The variable was coded 1 if respondents stated that they suffered from a maximum number of bleedings of more than 20 and 0 otherwise in the dataset. The third covariate (cov3) based on the question "How would you describe your current state of health in general?" It was coded 1 if respondents answered with very good and coded with 0 otherwise. Reference class is class 2. Respondents in class 1 differed significantly in terms of bleeding frequencies, maximum number of bleedings, and current state of health. Class 1 respondents

had a lower number of bleedings in the last year (coef. = 2.04; P < 0.05). Respondents of class 2 were more likely to have a maximum number of bleedings of more than 20 (coef. = -2.56; P < 0.01). Regarding current state of health, class 1 had a significant lower proportion of respondents with a very good health state (self-report by respondents) (coef. = -2.91; P < 0.01) than class 2 respondents.
(DOCX)

## Acknowledgments

We thank all participating patients and their families for their support. Our thanks to the support group "Deutschen Hämophiliegesellschaft zur Bekämpfung von Blutungskrankheiten e. V." (DHG e.V.), "IGH Interessengemeinschaft Hämophiler e.V." as well as the "HZRM Hämophilie-Zentrum Rhein Main", in particular Prof. Carmen Escuriola, for their help in recruiting the survey participants. Support for third-party editing assistance for this manuscript, furnished by Daniel Clyde, PhD, of Health Interactions, was provided by Roche Pharma AG, Grenzach-Wyhlen, Germany.

## Author Contributions

**Conceptualization:** Axel C. Mühlbacher, Björn Lamprecht.

**Data curation:** Axel C. Mühlbacher, Andrew Sadler, Björn Lamprecht.

**Formal analysis:** Axel C. Mühlbacher, Andrew Sadler, Björn Lamprecht.

**Funding acquisition:** Axel C. Mühlbacher, Björn Lamprecht.

**Investigation:** Axel C. Mühlbacher, Björn Lamprecht, Christin Juhnke.

**Methodology:** Axel C. Mühlbacher, Andrew Sadler.

**Project administration:** Axel C. Mühlbacher, Björn Lamprecht, Christin Juhnke.

**Resources:** Björn Lamprecht.

**Software:** Andrew Sadler.

**Supervision:** Axel C. Mühlbacher.

**Validation:** Axel C. Mühlbacher, Andrew Sadler, Björn Lamprecht, Christin Juhnke.

**Visualization:** Andrew Sadler, Christin Juhnke.

**Writing – original draft:** Axel C. Mühlbacher, Andrew Sadler, Christin Juhnke.

**Writing – review & editing:** Axel C. Mühlbacher, Andrew Sadler, Björn Lamprecht, Christin Juhnke.

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
