## [Decision Letter · Decision Letter 0]

9 Apr 2021

PONE-D-20-34726

Patient preferences in the treatment of hemophilia A: A Latent Class analysis

PLOS ONE

Dear Dr. Mühlbacher,

Thank you for submitting your manuscript to PLOS ONE. After careful consideration, we feel that it has merit but does not fully meet PLOS ONE’s publication criteria as it currently stands. Therefore, we invite you to submit a revised version of the manuscript that addresses the points raised during the review process.

Please note that while you will not see a long list of reviewer comments, a major revisions decision was appropriate given that the work involved in addressing them appeared to be a heavy lift. Each of the comments appears to be reasonable from my perspective, and if addressed appropriately, would improve the paper. Please be sure to address each carefully. 

We look forward to receiving your revised manuscript.

Kind regards,

Damian Adams

Academic Editor

PLOS ONE

Additional Editor Comments:

The paper has received two reviews, both encouraging revision and resubmission. Based on their comments, I believe a major revisions decision is appropriate, and would like to see the authors resubmit after carefully addressing each of the reviewers' comments and concerns, which appear reasonable. Upon resubmitting, please be sure to provide a detailed indication of how each comment and concern was addressed, either indicating changes made to the paper or otherwise explaining your approach. Since this is a major revisions decision, please note that the resubmitted manuscript will likely need to be assessed by the reviewers again. Thank you.

Journal Requirements:

2. Thank you for providing the following Funding Statement: 

"This study was financed by Roche Pharma AG (https://www.roche.de). Employees of the sponsor are listed as authors and were involved in the study design, data collection and analysis, decision to publish, or preparation of the manuscript.

All authors received support for third-party editing assistance, provided by Roche Pharma AG, Grenzach-Wyhlen, Germany. Mr. Sadler (M.Sc.) reports grants from Roche Pharma AG, Grenzach-Wyhlen, Germany, during the conduct of the study. Prof. Dr. Mühlbacher reports grants from Roche Pharma AG, Grenzach-Wyhlen, Germany, during the conduct of the study. Dr. Lamprecht is an employee of Roche Pharma AG, Grenzach-Wyhlen, Germany. Ms. Juhnke reports grants from Roche Pharma AG, Grenzach-Wyhlen, Germany, during the conduct of the study."

We note that one or more of the authors is affiliated with the funding organization, indicating the funder may have had some role in the design, data collection, analysis or preparation of your manuscript for publication; in other words, the funder played an indirect role through the participation of the co-authors.

If the funding organization did not play a role in the study design, data collection and analysis, decision to publish, or preparation of the manuscript and only provided financial support in the form of authors' salaries and/or research materials, please review your statements relating to the author contributions, and ensure you have specifically and accurately indicated the role(s) that these authors had in your study in the Author Contributions section of the online submission form. Please make any necessary amendments directly within this section of the online submission form.  Please also update your Funding Statement to include the following statement: “The funder provided support in the form of salaries for authors [insert relevant initials], but did not have any additional role in the study design, data collection and analysis, decision to publish, or preparation of the manuscript. The specific roles of these authors are articulated in the ‘author contributions’ section.”

If the funding organization did have an additional role, please state and explain that role within your Funding Statement.

Please also provide an updated Competing Interests Statement declaring this commercial affiliation along with any other relevant declarations relating to employment, consultancy, patents, products in development, or marketed products, etc.  

"All authors received support for third-party editing assistance, provided by Roche Pharma AG, Grenzach-Wyhlen, Germany. Mr. Sadler (M.Sc.) reports grants from Roche Pharma AG, Grenzach-Wyhlen, Germany, during the conduct of the study. Prof. Dr. Mühlbacher reports grants from Roche Pharma AG, Grenzach-Wyhlen, Germany, during the conduct of the study. Dr. Lamprecht is an employee of Roche Pharma AG, Grenzach-Wyhlen, Germany. Ms. Juhnke reports grants from Roche Pharma AG, Grenzach-Wyhlen, Germany, during the conduct of the study."

We note that one or more of the authors are employed by a commercial company: Gesellschaft für empirische Beratung GmbH.

3.1. Please provide an amended Funding Statement declaring this commercial affiliation, as well as a statement regarding the Role of Funders in your study. If the funding organization did not play a role in the study design, data collection and analysis, decision to publish, or preparation of the manuscript and only provided financial support in the form of authors' salaries and/or research materials, please review your statements relating to the author contributions, and ensure you have specifically and accurately indicated the role(s) that these authors had in your study. You can update author roles in the Author Contributions section of the online submission form.

3.2. Please also provide an updated Competing Interests Statement declaring this commercial affiliation along with any other relevant declarations relating to employment, consultancy, patents, products in development, or marketed products, etc.  

4. We noted in your submission details that a portion of your manuscript may have been presented or published elsewhere.

"Overall experimental design (Muhlbacher et al, Value Health, 2020).

This does not constitute dual publication as in the first paper we analyzed patients’ preferences of a whole sample set regarding general relative importance of all attributes with a mixed logit model (Muhlbacher et al, Value in Health, 2020; see uploaded paper as related manuscript file). Here, we aimed to assess heterogeneity of patients' preferences for alternative hemophilia A treatments in Germany; the main focus being to analyze possible differences in preference patterns in the sample regarding treatment characteristics."

Reviewers' comments:

Reviewer's Responses to Questions

**Comments to the Author**

1. Is the manuscript technically sound, and do the data support the conclusions?

Reviewer #1: Yes

Reviewer #2: Yes

2. Has the statistical analysis been performed appropriately and rigorously? 

Reviewer #1: Yes

Reviewer #2: Yes

3. Have the authors made all data underlying the findings in their manuscript fully available?

Reviewer #1: No

Reviewer #2: No

4. Is the manuscript presented in an intelligible fashion and written in standard English?

Reviewer #1: Yes

Reviewer #2: Yes

5. Review Comments to the Author

Reviewer #1: This brief paper describes a latent class analysis on BWS case-3 responses regarding homophilia A treatment preferences of 57 respondents. Unlike their primary analysis, this paper describes two classes. This paper represents a modest contribution based on the topic and interpretation. It does perform above the basics or introduce innovations in methods, practice, or application. As an exploratory analysis,

Over half of this secondary paper re-iterates description found in the primary one. Its primary contribution is the application of the stata LCM package to examine heterogeneity. The authors provide a modest interpretation of the grade of membership and differences in attribute importance. They did not include any qualitative evidence to bolster the explanation and did not conduct a confirmatory analysis.

Major comments:

1. Please describe the respondents in detail, including aspects of their recruitment, characteristics, and interview experience, which may affect their grade-of-membership and the implications of the results.

2. Please include screenshots of the full survey instrument. These screenshots are typically mandatory prior to the review of any stated preference evidence.

3. Please emphasize the difference between taste and scale in the interpretation of LCM results.

Reviewer #2: The author conducts a latent class analysis for identifying hidden classes for patient preferences for hemophilia A. The research question is an interesting and important one. But i have some methodological queries. The author has failed to mention how their results will be used to identify hidden cohorts in the real world. When we conduct a latent class analysis, we identify hidden classes. It is to be understood by varying the variables involved in latent class model formation; we will identify different classes. So after having the rationality for the selection of questions, a latent model needs to be formed. (until here, the author has conducted though rationality for final question utilized can be elaborated further from a clinical and statistical standpoint). After the hidden population has been identified, they need to extrapolate the identified class variable (either 0 or 1) to the primary data frame and make a univariate analysis (on the unused variables) to distinguish between the identified population. The author also mentions that there seems to be no difference in demographics between the latent class. How will we use it in the real world to differentiate between the hidden class so that the healthcare worker can plan for interventions to address the issues? I want the author to address this critical question.

The author need to elaborate the discussion. The discussion is too short. Elaborate on past studies and papers which have adopted similar methodology in other diseases.

In the introduction, the author has written prevalence in the form of mean and sd.

Is the first paper, which the author referring to also utilizing the same data. The authors should make sure that there is no result repetition.

By means of not explaining the variable distribution to the study population, it would be difficult for readers to understand.

The writeup is more statistically oriented concentrating on non-important attributes. I would ask the author to have a thorough write-up overhaul having in mind that PlosOne has a broad readership. Also try to explain the statistical concepts then and there whereever you are mentioning.Explaining a tough concept lucidly to the reader is also an important art.

6. PLOS authors have the option to publish the peer review history of their article (what does this mean?). If published, this will include your full peer review and any attached files.

Reviewer #1: No

Reviewer #2: **Yes: **Praveen Kumar M

---

## [Author Response · Author response to Decision Letter 0]

14 Jun 2021

Dear Dr. Adams,

Dear Reviewers,

On behalf of my co-authors, thank you for providing us with the opportunity to revise our manuscript for PLOS ONE. As requested, please find a point-by-point response to each of the reviewers’ queries and suggestions in the regarding uploaded document, as well as the items mentioned in the compliance form.

Thank you once again for your time; I hope to hear from you in due course.

Kind regards,

Axel C. Mühlbacher

---

## [Decision Letter · Decision Letter 1]

10 Aug 2021

Patient preferences in the treatment of hemophilia A: A Latent Class analysis

PONE-D-20-34726R1

Dear Dr. Mühlbacher,

We’re pleased to inform you that your manuscript has been judged scientifically suitable for publication and will be formally accepted for publication once it meets all outstanding technical requirements.

Kind regards,

Damian Adams

Academic Editor

PLOS ONE

Reviewers' comments:

Reviewer's Responses to Questions

**Comments to the Author**

1. If the authors have adequately addressed your comments raised in a previous round of review and you feel that this manuscript is now acceptable for publication, you may indicate that here to bypass the “Comments to the Author” section, enter your conflict of interest statement in the “Confidential to Editor” section, and submit your "Accept" recommendation.

Reviewer #1: All comments have been addressed

Reviewer #2: All comments have been addressed

2. Is the manuscript technically sound, and do the data support the conclusions?

Reviewer #1: (No Response)

Reviewer #2: Yes

3. Has the statistical analysis been performed appropriately and rigorously? 

Reviewer #1: (No Response)

Reviewer #2: Yes

4. Have the authors made all data underlying the findings in their manuscript fully available?

Reviewer #1: (No Response)

Reviewer #2: No

5. Is the manuscript presented in an intelligible fashion and written in standard English?

Reviewer #1: (No Response)

Reviewer #2: Yes

6. Review Comments to the Author

Reviewer #1: Congrats!Congrats!Congrats!Congrats!Congrats!Congrats!Congrats!Congrats!Congrats!Congrats!Congrats!Congrats!

Reviewer #2: The author team has addressed all the queries raised to a satisfactory extent. Congrats to the team for coming up with the revision. Thanks.

7. PLOS authors have the option to publish the peer review history of their article (what does this mean?). If published, this will include your full peer review and any attached files.

Reviewer #1: No

Reviewer #2: **Yes: **Praveen Kumar M

---

## [Editor Report · Acceptance letter]

13 Aug 2021

PONE-D-20-34726R1 

Patient preferences in the treatment of hemophilia A: A Latent Class analysis 

Dear Dr. Mühlbacher:

I'm pleased to inform you that your manuscript has been deemed suitable for publication in PLOS ONE. Congratulations! Your manuscript is now with our production department. 

Kind regards, 

on behalf of

Dr. Damian Adams 

Academic Editor

PLOS ONE